# TEM,CTX-M,SHV Genes in ESBL-Producing *Escherichia coli* and *Klebsiella pneumoniae* Isolated from Clinical Samples in a County Clinical Emergency Hospital Romania-Predominance of CTX-M-15

**DOI:** 10.3390/antibiotics11040503

**Published:** 2022-04-10

**Authors:** Alice Elena Ghenea, Ovidiu Mircea Zlatian, Oana Mariana Cristea, Anca Ungureanu, Radu Razvan Mititelu, Andrei Theodor Balasoiu, Corina Maria Vasile, Alex-Ioan Salan, Daniel Iliuta, Mihaela Popescu, Anca-Loredana Udriștoiu, Maria Balasoiu

**Affiliations:** 1Department of Bacteriology-Virology-Parasitology, University of Medicine and Pharmacy of Craiova, 200349 Craiova, Romania; gaman_alice@yahoo.com (A.E.G.); ovidiu.zlatian@gmail.com (O.M.Z.); ancaungureanu65@yahoo.com (A.U.); mradurazvan@gmail.com (R.R.M.); balasoiu.maria@yahoo.com (M.B.); 2Department of Ophthalmology, University of Medicine and Pharmacy of Craiova, 200349 Craiova, Romania; andrei_theo@yahoo.com; 3Department of Paediatrics, University of Medicine and Pharmacy of Craiova, 200349 Craiova, Romania; 4Department of Oral and Maxillofacial Surgery, University of Medicine and Pharmacy of Craiova, 200349 Craiova, Romania; alex.salan@umfcv.ro; 5Department of Psychiatry, University of Medicine and Pharmacy of Craiova, 200349 Craiova, Romania; iliutadaniel94@gmail.com; 6Department of Endocrinology, University of Medicine and Pharmacy of Craiova, 200349 Craiova, Romania; mihaela.n.popescu99@gmail.com; 7Faculty of Automation, Computers and Electronics, University of Craiova, 200776 Craiova, Romania; anca.udristoiu@edu.ucv.ro

**Keywords:** antimicrobial resistance, *Klebsiella pneumoniae*, *Escherichia coli*, CTX-M-15, ESBL, antibiotic resistance

## Abstract

Background: CTX-M betalactamases have shown a rapid spread in the recent years among Enterobacteriaceae and have become the most prevalent Extended Spectrum Beta-Lactamases (ESBLs) in many parts of the world. The introduction and dissemination of antibiotic-resistant genes limits options for treatment, increases mortality and morbidity in patients, and leads to longer hospitalization and expensive costs. We aimed to identify the beta-lactamases circulating encoded by the genes *bla*_CTX-M-15_, *bla*_SHV-1_ and *bla*_TEM-1_ in *Escherichia coli (E. coli)* and *Klebsiella pneumoniae (K. pneumoniae)* strains. Furthermore, we established the associated resistance phenotypes among patients hospitalized in the Intensive Care Unit (ICU) from County Clinical Emergency Hospital of Craiova, Romania. Methods: A total of 46 non-duplicated bacterial strains (14 strains of *E. coli* and 32 strains of *K. pneumoniae*), which were resistant to ceftazidime (CAZ) and cefotaxime (CTX) by Kirby–Bauer disk diffusion method, were identified using the automated VITEK2 system. Detection of ESBL-encoding genes and other resistance genes was carried out by PCR. Results. *E. coli* strains were resistant to 3rd generation cephalosporins and moderately resistant to quinolones, whereas *K. pneumoniae* strains were resistant to penicillins, cephalosporins, and sulfamides, and moderately resistant to quinolones and carbapenems. Most *E. coli* strains harbored *bla*_CTX-M-15_ gene (13/14 strains), a single strain had the *bla*_SHV-1_ gene, but 11 strains harbored *bla*_TEM-1_ gene. The *mcr-1* gene was not detected. We detected *tet*(A) gene in six strains and *tet*(B) in one strain. In *K. pneumoniae* strains we detected *bla*_CTX-M-15_ in 23 strains, *bla*_SHV_-_1_ in all strains and *bla*_TEM-1_ in 14 strains. The colistin resistance gene *mcr-1* was not detected. The tetracycline gene *tet*(A) was detected in 11 strains, but the gene *tet*(B) was not detected in any strains. Conclusions. The development in antibiotic resistance highlights the importance of establishing policies to reduce antibiotic use and improving the national resistance surveillance system in order to create local antibiotic therapy guidelines.

## 1. Introduction

Enterobacteria-producing extended-spectrum beta-lactamases (ESBL) play an important role in increasing hospitalization time, healthcare infections, mortality, and morbidity rates. [1]. *Escherichia coli (E.coli)* is a commensal of the intestinal flora, but may be involved in diseases such as septicemia, urinary tract infections, or purulent infections, in which the capacity to acquire antibiotic resistance genes is obviously increased [2,3]. *Klebsiella pneumoniae* (*K. pneumoniae*) can also be found in the intestinal flora, being an opportunistic pathogen involved in the production of nosocomial infections, especially in immunocompromised patients and hospitalized in the intensive care unit. This pathogen can also acquire resistance genes to different classes of antibiotics, which underlie the increased pathogen potential and the appearance of severe forms of the disease [2,4,5,6]. 

The most common mechanism of beta-lactam antibiotic resistance is beta-lactamase production. Due to extended-spectrum beta-lactamases (ESBLs), this phenomenon of resistance has increased in medical practice [7,8,9,10]. They are plasmid-encoded enzymes with the ability to hydrolyze bonds of β-lactam rings from antibiotics like penicillins, cephalosporins, and aztreonam, and are inhibited by the action of clavulanic acid [8,9]. Due to the coexistence of different modifying enzymes on the same plasmid, there is a possibility of resistance to other classes of antibiotics, such as fluoroquinolones, aminoglycosides, tetracyclines, and trimethoprim sulfamethoxazole [11,12]. 

In various studies, the most investigated types are TEM, SHV, OXA, and CTX-M of these, CTX-M being the most frequently associated with resistance to these antibiotics [10,11,12]. There are more than 130 different types of CTX-M enzymes divided into five different groups as follows: CTX-M-1, CTX-M-2, CTX-M-8, CTX-M-9, and CTX-M-25 [13]. Of these, the most common type in hospitals and in the community is the CTX-M-15 type [13]. Currently, more than 140 different types of TEM type beta-lactamases are known, TEM-1 being most encountered in *E. coli* and *K. pneumoniae*. There are also more than 100 different types of SHV, especially existent in *Pseudomonas aeruginosa* and *Acinetobacter* spp. [14].

The emergence and spread of these antibiotic-resistant genes reduce the chances of treatment, and increase the mortality and morbidity of patients with prolonged hospitalization and high costs [15]. Therefore, it is necessary to identify them as soon as possible and apply measures to reduce the prevalence of ESBLs. It is a priority to treat patients with effective antimicrobials, to maintain proper hygiene rules, and to try as far as possible to avoid certain invasive procedures, such as central venous catheterization [14].

For detection of multidrug-resistance, often the conventional antimicrobial testing is not sufficient [16]. Often, microorganisms are not identified as ESBL producers and so they can spread in the hospital environment [17] so the identification of resistant phenotypes is crucial, especially in countries with excessive use of antibiotics and deficient infection control measures [18,19]. 

The main objective of this work was to establish the beta-lactamases circulating in the Oltenia region of Romania, encoded by the genes *bla*_CTX-M-15_, *bla*_SHV-1,_ and *bla*_TEM-1_ in *E. coli* and *K. pneumoniae* strains and also the associated resistance phenotypes among patients from the County Clinical Emergency Hospital of Craiova, Romania.

## 2. Material and Methods 

### 2.1. Bacterial Strains

We collected 46 non-duplicated bacterial strains (14 strains of *E. coli* and 32 strains of *K. pneumoniae*), which were resistant to ceftazidime (CAZ) and cefotaxime (CTX) by the diffusimetric Kirby–Bauer method, from patients hospitalized in the Intensive Care Unit (ICU) of County Clinical Emergency Hospital of Craiova, Romania, which serves patients from the entire Oltenia region, between 1 January 2020–31 December 2021. Strains were obtained from purulent secretions, catheters, peritoneal fluids, tracheal aspirates, and sputum. 

### 2.2. Antimicrobial Susceptibility Testing

Antimicrobial susceptibility testing was accomplished by Kirby–Bauer disk diffusion method, for ceftazidime (30 µg) and cefotaxime (30 µg), according to the standard Clinical Laboratory Standards Institute (CLSI, 2019) [20]. *E. coli* and *K. pneumoniae* strains that were resistant to both antibiotics were tested using the automated VITEK2 system (Biomerieux, Marcy-l’Étoile, France) against amikacin, ampicillin, aztreonam, cefazolin, cefepime, ceftriaxone, cefuroxime, ciprofloxacin, tetracycline, gentamycin, tobramycin moxifloxacin, norfloxacin, piperacillin/tazobactam, sulphamethoxazole/trimethoprim, tigecycline, tobramycin, imipenem, meropenem, and ertapenem. The internal quality control for Muller-Hinton agar and antibiotic disks was performed using control strains *E. coli* ATCC 25922, *E. coli* (BAA-197) and *K. pneumoniae* (ATCC 51503, ATCC 700603).

The confirmation of ESBL production was carried out phenotypically using a double-disk synergy test [20] with kits from ROSCO (Taastrup, Denmark).

A strain was classified as multidrug-resistant (MDR) if it was resistant to at least one agent in three or more antimicrobial classes [21]. 

### 2.3. Molecular Testing

Genomic DNA was extracted from *E. coli* and *K. pneumoniae* strains as follows. Colonies were suspended in 184 mL tube with sterile water and we added 16 mL proteinase K (15 mg/mL). The mixture was incubated for 30 min at 56 °C. Automated DNA extraction was then performed using Maxwell^®^ 16 System (Promega, Madison, WI, USA) instrument with the extraction kit Maxwell^®^ 16 Cell DNA Purification Kit (Promega, USA) that is suitable for both eukaryotic and prokaryotic DNA extraction.

The extended spectrum beta-lactamase genes (*bla*_CTX-M-15_, *bla*_SHV-1_ and *bla*_TEM-1_) were detected through in-house PCR using consensus primers and methodology according to those described by Monstein et al. [22]. The multiplex PCR assay was carried out using GoTaq^®^ Poymerase and dNTP Mix from Promega manufacturer on an Eppendorf MasterCycler^®^ thermal cycler. The primers for bla_CTX_/bla_SHV_/bla_TEM_ genes were ordered from Invitrogen—“Class A extended beta-lactamase primers”. Electrophoretic separation of resulted amplicons was done using ethidium bromide-stained 1% agarose gel from Sigma Aldrich. We also screened the strains for the presence of the *mcr*-1 gene [23] and the *tet*(A)/*tet*(B) genes [24] using single PCR assay with similar parameters to those of Adelowo and Fagade [24]. Amplicon detection and evaluation from agarose gels for each of the techniques we employed was done on a G:Box Chemi HR (Syngene) Bio Imaging system. Standard *E. coli* (BAA-197) and *K. pneumoniae* (ATCC 51503, ATCC 700603) strains producing ESBLs, as well as DNA from previous extractions from known enzymes producing strains were used as controls.

### 2.4. Statistical Analysis

The data collected from the informatic system for all microbiology samples were introduced on a weekly basis into the software STATA (StataCorp LLC, TX, USA), which calculated the resistance profile and the classification in MDR or non-MDR criteria according to ECDC guidelines algorithm [21], in order to select the strains to be included in the study. We also used the software to generate the tables and figures presented in this study. 

## 3. Results

All the strains being studied fulfilled the criteria for MDR [21] and were positive for phenotypic ESBL test. 

### 3.1. Antibiotic Resistance in E. coli

All 14 strains of *E. coli* were resistant to ampicillin, cefazolin, ceftazidime, ceftriaxone, and cefuroxime as in Figure 1.

A high rate of resistance was observed for cefepime (*n* = 12; 85.17%) and aztreonam (*n* = 10; 71.43%). Relative moderate resistance was observed for tetracycline, gentamycin, and tobramycin (*n* = 8; 57.14%); moxifloxacin (*n* = 7; 50.00%); ciprofloxacin (*n* = 6; 42.86%); norfloxacin (*n* = 6; 42.86%); and piperacillin/tazobactam (*n* = 16; 50.00%). Relatively low resistance rates under 30% were observed for amikacin, tigecycline, and meropenem (*n* = 2; 14.29%), and also for piperacillin/tazobactam, sulfamethoxazole/trimethoprim, imipenem, and ertapenem (*n* = 4; 28.57%). All strains were susceptible to colistin.

### 3.2. Genetic Characteristics of Multidrug-Resistant E. coli Strains

Many of the 14 *E. coli* strains harbored *bla*_CTX-M-15_ gene (13 strains), most from purulent secretions. A single *E. coli* strain had the *bla*_SHV-1_ gene, but 11 strains harbored *bla*_TEM-1_ gene. The *mcr-1* gene was not detected We detected *tet*(A) gene in six strains and *tet*(B) in one strain.

### 3.3. Antibiotic Resistance in K. pneumoniae

All 32 strains of *K. pneumoniae* were resistant to ampicillin, cefazolin, ceftazidime, cefuroxime, and cefotaxime. Twenty-six strains were resistant to ceftriaxone and cefepime (81.25%). From the carbapenem group 20 strains were resistant to imipenem (62.50%), 16 to ertapenem (50.00%), and 14 to meropenem (43.75%). Besides beta-lactam antibiotics, 14 strains were resistant to gentamycin and tetracycline, 12 to tigecycline, 16 to piperacillin/tazobactam (50.00%), and 24 to sulfamethoxazole/trimethoprim (75.00%). The resistance to quinolones was also important, as 22 strains were resistant to ciprofloxacin (68.75%), 12 to moxifloxacin (37.50%), and 20 to norfloxacin (62.50%). Very low resistance was recorded to amikacin (4 strains: 12.5%). All strains were susceptible to colistin (Figure 2).

### 3.4. Genetic Characteristics of the K. pneumoniae Strains

In *K. pneumoniae* strains, we detected beta-lactamase genes *bla*_CTX-M-15_ in 23 strains, *bla*_SHV_-_1_ in all strains, and *bla*_TEM-1_ in 14 strains. The colistin resistance gene *mcr-1* was not detected. The tetracycline gene *tet*(A) was detected in 11 strains, but the gene *tet*(B) was not detected in any strains.

## 4. Discussion

Frequently, beta-lactam antibiotics are used as the first-line drugs in infections caused by *Enterobacteriaceae* [25]. This study investigates the ESBL-producing *Enterobacteriaceae* in a tertiary care Romanian hospital. We confirmed the spread of the gene *bla*_CTX-M-15_ in isolates from Romania and that these strains show resistance to various antibiotic classes. 

The prevalence of ESBL-producing *Enterobacteriaceae* varies greatly around the globe. In all countries, in settings where the patients are treated with antibiotics, there is a high prevalence of multidrug-resistant *Enterobacteriaceae* [26]. Low rates were reported in North America and Europe [27,28] and high rates in Asia [29], South America [30], and Africa [31]. Their spread is encouraged by antimicrobials used without a prescription, poor hygiene, counterfeit drugs, high prevalence of infectious diseases, and lack of diagnostic tools for infections and antibiotic resistances [32,33]. 

The rates of resistance of *Enterobacteriaceae* nosocomial isolates are increasing also in Romania [34,35,36,37] as in other countries [38,39,40,41,42]. There are insufficient studies on the nature and prevalence of the ESBLs in nosocomial settings of Romania. In this study, we describe the prevalence and nature of the ESBLs produced by the *Enterobacteriaceae* circulating in a hospital from the Oltenia region of Romania. 

In the ICU, patients with systemic infections are treated with a massive number of antibiotics [43]. These strains are probably transmitted through the air in healthcare settings, particularly in the ICU between patients on mechanical ventilation [44]. This transmission route is supported by the fact that 4 out of 14 *E. coli* strains and 8 out of the 32 *Klebsiella pneumoniae* strains were recovered from the samples from the respiratory tract.

The penicillin and cephalosporin resistance patterns observed in our strains can be explained by the excessive use of ceftriaxone and cefotaxime in the empirical therapy in hospitals, and hospitalization itself has been identified as a risk factor for infection with ESBL-producing *Enterobacteriaceae* as plasmids carrying genes that encode ESBLs can be easily transmitted horizontally between different bacteria in the hospital environment [45,46,47]. The study shows an association of the ESBL production with resistance to aminoglycosides (excepting amikacin), tetracycline, and trimethoprim/sulfamethoxazole, as previously reported [48,49,50]. The cross-resistance phenomenon is alarming because it restricts greatly the choices for empirical therapy in patients infected with ESBL-producing *Enterobacteriaceae*. 

In our study, low resistance rates were observed to amikacin, tigecycline, and meropenem in *E. coli* and *K. pneumoniae* strains that had moderate resistance to the same antibiotics. However, meropenem should be used with caution in empirical therapy to avoid the selection of carbapenemases-producing *Enterobacteriaceae* and amikacin is toxic for kidneys. Nevertheless, in serious infections with ESBL-producing *Enterobacteriaceae*, the carbapenems are currently considered drugs of choice, for example, ertapenem once daily. Sometimes h the patients show a partial response to non-carbapenem antibiotics [51,52]. Some authors suggested that amikacin can be successfully used in these infections due to the low resistance rates in *Enterobacteriaceae* [53]. 

There is a strong geographical variation in the CTX-M type of ESBLs in *E. coli* and *K. pneumoniae* isolates. The most prevalent allele with a worldwide distribution is CTX-M-15, which belongs to the CTX-M-1 group [8,38,54]. CTX-M-14 is highly prevalent in some Asian countries [55], whereas in South America predominates CTX-M-2 and CTX-M-8 [55,56]. 

We found the *bla*_CTX-M-15_ in more than 90% of the *E. coli* isolates (Table 1), a finding that resembles other studies in nosocomial settings in Portugal [57], Lithuania [58], Hungary [59], and Germany [60], where it was the most prevalent ESBL type. Recently, blaCTX-M-15 was detected in *E. coli* in cats and dogs [61,62], other companion animals and in poultry [63]. Similar results were reported from Spain [64]. CTX-M-15 is frequently associated with the epidemic clone ST131 with worldwide distribution [62,65,66]. In our study, the prevalence of the *bla*_CTX-M-15_ gene was detected in 13/14 strains in *E. coli* (Table 1) and 23/32 strains in *Klebsiella pneumoniae* (Table 2); this high prevalence was previously reported in samples obtained from humans [4,61,67,68] and pets in Portugal [69], Italy [70], and Germany [60]. The beta-lactamase CTX-M-15 is endemic in various countries and rapidly spreads to different *Enterobacteriaceae* species [14]. In European countries, a high prevalence of CTX-M-15 has been reported in many studies [58,60,61,63]. These data show a spread of *bla*_CTX-M-15_ gene in both humans and animals across Europe.

In Romania, there are a few studies on the prevalence of CTXM-15 ESBL variant in isolates from clinical settings [71,72,73] and in pets [63,74,75].

We had a few strains of *E. coli* and *K. pneumoniae* in which CTXM-15 gene was not detected but they were resistant to 3rd generation cephalosporins. That can be explained by the presence of other beta-lactamases, as carbapenemases (because these strains were resistant to carbapenems). Indeed, the KPC (*K. pneumoniae* carbapenemases) [76] and NDM (New Delhi Metallobetalactamases) lactamases [77] can induce resistance not only to carbapenems, but also to cephalosporins.

The *bla*_SHV-1_ gene was detected in our study in only one *E. coli* strain from 14 studied, but in all *K. pneumoniae* strains, in accordance with other authors, who found it sometimes in association with *ctx* genes, in both animals and humans [6,62]. The ESBLs SHV have also been detected in animal wildlife [78,79,80].

Regarding the TEM beta-lactamases, in our study, they have been detected in large proportion in *E. coli* strains (11/14 strains) and *K. pneumoniae* strains (14/32 strains). Indeed, the *bla*_TEM-1_ genes were detected more and more frequently in Italian hospitals [81] since new *tem* variants were first detected in *K. pneumoniae* in France [82] and Korea [83] in 1998.

The prevalence of TEM beta-lactamases was associated with the usage of moxalactam. This drug is not used in Romania, therefore, we suppose that the selective pressure was due to cefotaxime, a commonly used drug. This is supported by the evidence that some TEM beta-lactamases hydrolase easily the cefotaxime [82]. 

The TEM- and SHV-type ESBL distribution found in our study was different, overall, from that observed in other countries, confirming a notable variability in the epidemiology of these resistance determinants and underscoring the need to monitor them in different epidemiological settings. It would be interesting to investigate the genetic support and transferability of the various ESBL determinants and the clonal diversity of isolates to understand the role of vector versus clonal spread in the dissemination of these resistance genes.

## 5. Limitations

One limitation of this study is that we had not known the previous antibiotic treatments in the patients prior to sample collection that could have induced the selection of multidrug-resistant strains. 

Another limitation is that we only use primers for specific alleles of the *bla*_CTX_, *bla*_SHV_, and *bla*_TEM_ genes, but is possible that the studied strains could also harbor other variants of these resistance genes that were not detected by our methodology. 

## 6. Conclusions

This report reveals a high antibiotic resistance in ESBL-producing *Enterobacteriaceae* circulating in the Oltenia region of Romania and the predominance of the CTXM-15 enzyme among those, especially in *E. coli*. The SHV enzyme was scarce in *E. coli* but present in all *K. pneumoniae strains*, and the TEM enzyme was more frequent in *E. coli* than in *K. pneumoniae*. The increased resistance emphasizes the need to implement policies of rationalizing antibiotic use and strengthen the national resistance surveillance system in order to develop local antibiotic therapy guidelines.

## Figures and Tables

**Figure 1 antibiotics-11-00503-f001:**
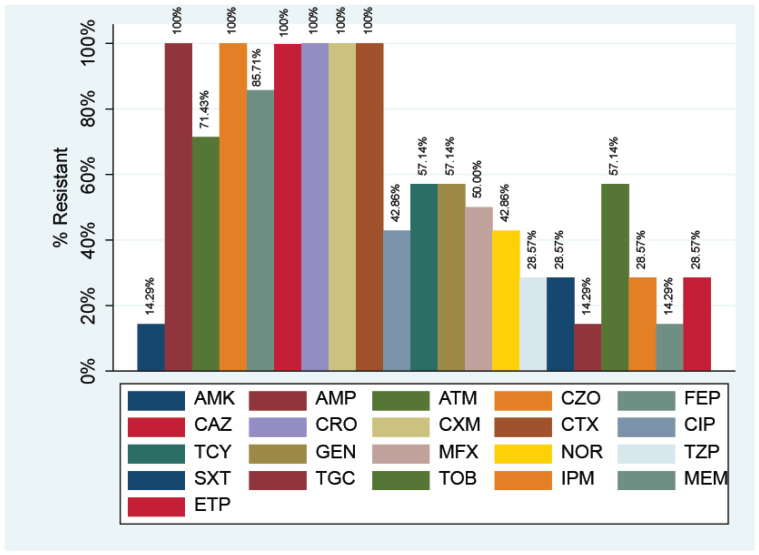
Resistance of *E. coli* strains to various antibiotics. (AMP: Ampicillin; AMK: Amikacin; ATM: Aztreonam; CZO: Cephazolin; FEP: Cephepime; CAZ: Cephtazidime; CRO: Ceftriaxiaxone; CXM: Cefuroxime; CTX: Cefotaxime; TGC: Tigecyclin; TPZ: Piperacillin/Tazobactam; MEM: Meropenem; CIP: Ciprofloxacin; SXT: Sulphametoxazole/Trimethoprim; ETP: Ertapenem; GEN: Gentamycin; MXF: Moxifloxacin; NOR: Norfloxacin; TCY: Tetracycline; TOB: Tobramycin).

**Figure 2 antibiotics-11-00503-f002:**
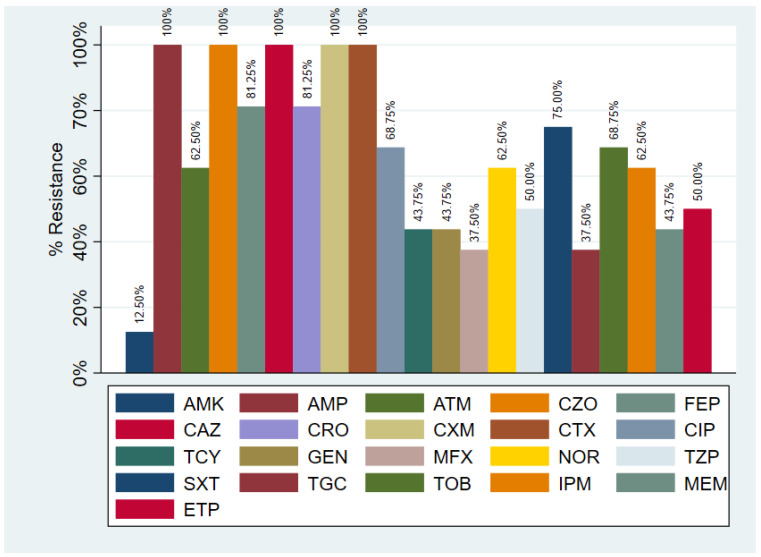
Resistance of *K. pneumoniae* strains to various antibiotics. (AMP: Ampicillin; AMK: Amikacin; ATM: Aztreonam; CZO: Cephazolin; FEP: Cephepime; CAZ: Cephtazidime; CRO: Ceftriaxone; CXM: Cefuroxime; CTX: Cefotaxime; TGC: Tigecyclin; TPZ: Piperacillin/Tazobactam; MEM: Meropenem; CIP: Ciprofloxacin; SXT: Sulphametoxazole/Trimethoprim; ETP: Ertapenem; GEN: Gentamycin; MXF: Moxifloxacin; NOR: Norfloxacin; TCY: Tetracycline; TOB: Tobramycin).

**Table 1 antibiotics-11-00503-t001:** Resistance phenotype and associated genotype in the 14 *E. coli* strains from clinical samples in a Romanian hospital.

No.	Origin	Resistance Phenotype	Beta-Lactamases	*tet* Genes
1	Purulent secretion	AMP ATM CZO FEP CAZ CRO CXM CTX	CTX-M-15 TEM-1	ND
2	Purulent secretion	AMP ATM CZO FEP CAZ CRO CXM CTX GEN TCY	CTX-M-15 TEM-1	ND
3	Purulent secretion	AMP ATM CZO FEP CAZ CRO CXM CTX CIP MFX NOR GEN TOB SXT TGC TCY	CTX-M-15 TEM-1	*tet*(A)
4	Purulent secretion	AMK AMP TZP ATM CZO FEP CAZ CRO CXM CTX CIP MFX NOR GEN TOB SXT IPM MEM ETP	CTX-M-15 TEM-1	ND
5	Tracheal aspirate	AMP CZO CAZ CRO CXM CTX MFX GEN TOB	CTX-M-15	ND
6	Tracheal aspirate	AMP ATM CZO FEP CAZ CRO CXM CTX GEN TOB TCY ETP	TEM-1	*tet*(A)
7	Catheter	AMP CZO FEP CAZ CRO CXM CTX CIP MFX NOR TOB TCY IPM	CTX-M-15 TEM-1	*tet*(A)
8	Purulent secretion	AMP ATM CZO FEP CAZ CRO CXM CTX	CTX-M-15 TEM-1	ND
9	Purulent secretion	AMP ATM CZO FEP CAZ CRO CXM CTX GEN TCY	CTX-M-15 TEM-1	*tet*(B)
10	Purulent secretion	AMK AMP TZP ATM CZO FEP CAZ CRO CXM CTX CIP MFX NOR GEN TOB SXT TGC TCY	CTX-M-15 TEM-1	*tet*(A)
11	Purulent secretion	AMP TZP ATM CZO FEP CAZ CRO CXM CTX CIP MFX NOR GEN TOB SXT IPM MEM ETP	CTX-M-15 SHV-1	ND
12	Tracheal aspirate	AMP CZO CAZ CRO CXM CTX NOR	CTX-M-15	ND
13	Tracheal aspirate	AMP TZP ATM CZO FEP CAZ CRO CXM CTX TCY ETP	CTX-M-15 TEM-1	*tet*(A)
14	Catheter	AMP CZO FEP CAZ CRO CXM CTX CIP MFX NOR TOB TCY IPM	CTX-M-15 TEM-1	*tet*(A)

**Table 2 antibiotics-11-00503-t002:** Resistance phenotype and associated genotype in the 32 *K. pneumoniae* strains from clinical samples in a Romanian hospital.

No.	Origin	Resistance Phenotype	Beta-Lactamases	*tet* Genes
1	Sputum	AMK AMP TZP ATM CZO FEP CAZ CRO CXM CTX CIP NOR GEN TOB SXT TGC TCY IPM MEM ETP	CTX-M-15 SHV-1 TEM-1	*tet*(A)
2	Sputum	AMP CZO FEP CAZ CRO CXM CTX CIP MFX NOR GEN TOB SXT TCY IPM MEM	CTX-M-15 SHV-1	*tet*(A)
3	Peritoneal fluid	AMP ATM CZO FEP CAZ CRO CXM CTX CIP MFX NOR SXT IPM	CTX-M-15 SHV-1 TEM-1	ND
4	Peritoneal fluid	AMP TZP ATM CZO FEP CAZ CRO CXM CTX CIP MFX NOR SXT TGC IPM MEM ETP	CTX-M-15 SHV-1 TEM-1	ND
5	Purulent secretion	AMP CZO CAZ CXM CTX NOR	SHV-1	ND
6	Purulent secretion	AMK AMP ATM CZO FEP CAZ CRO CXM CTX CIP NOR GEN TOB SXT TGC MEM ETP	CTX-M-15 SHV-1	ND
7	Purulent secretion	AMP CZO CAZ CXM CTX CIP TOB TCY	SHV-1	*tet*(A)
8	Purulent secretion	AMP CZO CAZ CXM CTX NOR TCY	SHV-1	*tet*(A)
9	Purulent secretion	AMP TZP ATM CZO FEP CAZ CRO CXM CTX SXT TCY	SHV-1	*tet*(A)
10	Purulent secretion	AMP TZP ATM CZO FEP CAZ CRO CXM CTX CIP TOB SXT TGC IPM MEM ETP	CTX-M-15 SHV-1	ND
11	Purulent secretion	AMP CZO FEP CAZ CRO CXM CTX CIP TOB SXT IPM	CTX-M-15 SHV-1	ND
12	Purulent secretion	AMP TZP CZO FEP CAZ CRO CXM CTX TOB	CTX-M-15 SHV-1	ND
13	Purulent secretion	AMP ATM CZO FEP CAZ CRO CXM CTX GEN TOB SXT TGC IPM ETP	CTX-M-15 SHV-1 TEM-1	ND
14	Tracheal aspirate	AMP TZP ATM CZO FEP CAZ CRO CXM CTX CIP MFX NOR GEN TOB SXT IPM ETP	CTX-M-15 SHV-1 TEM-1	ND
15	Tracheal aspirate	AMP TZP ATM CZO FEP CAZ CRO CXM CTX CIP MFX NOR GEN TOB SXT TGC TCY IPM MEM ETP	CTX-M-15 SHV-1 TEM-1	ND
16	Catheter	AMK AMP TZP ATM CZO FEP CAZ CRO CXM CTX CIP MFX NOR GEN TOB SXT TCY IPM MEM ETP	CTX-M-15 SHV-1 TEM-1	ND
17	Sputum	AMK AMP TZP ATM CZO FEP CAZ CRO CXM CTX CIP NOR GEN TOB SXT TGC TCY IPM MEM ETP	CTX-M-15 SHV-1 TEM-1	ND
18	Sputum	AMP CZO FEP CAZ CRO CXM CTX CIP MFX NOR GEN TOB SXT TCY IPM MEM	CTX-M-15 SHV-1	*tet*(A)
19	Peritoneal fluid	AMP ATM CZO FEP CAZ CRO CXM CTX CIP MFX NOR SXT IPM	CTX-M-15 SHV-1 TEM-1	ND
20	Peritoneal fluid	AMP TZP ATM CZO FEP CAZ CRO CXM CTX CIP MFX NOR SXT TGC IPM MEM ETP	CTX-M-15 SHV-1 TEM-1	ND
21	Wound secretion	AMP CZO CAZ CXM CTX NOR	SHV-1	ND
22	Wound secretion	AMP ATM CZO FEP CAZ CRO CXM CTX CIP NOR GEN TOB SXT TGC MEM ETP	CTX-M-15 SHV-1	ND
23	Wound secretion	AMP CZO CAZ CXM CTX CIP TOB TCY	SHV-1	*tet*(A)
24	Wound secretion	AMP CZO CAZ CXM CTX NOR TCY	SHV-1	*tet*(A)
25	Wound secretion	AMP TZP ATM CZO FEP CAZ CRO CXM CTX SXT TCY	SHV-1	*tet*(A)
26	Wound secretion	AMP TZP ATM CZO FEP CAZ CRO CXM CTX CIP TOB SXT TGC IPM MEM ETP	CTX-M-15 SHV-1	ND
27	Wound secretion	AMP CZO FEP CAZ CRO CXM CTX CIP TOB SXT IPM	CTX-M-15 SHV-1	ND
28	Wound secretion	AMP TZP CZO FEP CAZ CRO CXM CTX TOB	SHV-1	ND
29	Wound secretion	AMP ATM CZO FEP CAZ CRO CXM CTX GEN TOB SXT TGC IPM ETP	CTX-M-15 SHV-1 TEM-1	ND
30	Tracheal aspirate	AMP TZP ATM CZO FEP CAZ CRO CXM CTX CIP MFX NOR GEN TOB SXT IPM ETP	CTX-M-15 SHV-1 TEM-1	ND
31	Tracheal aspirate	AMP TZP ATM CZO FEP CAZ CRO CXM CTX CIP MFX NOR GEN TOB SXT TGC TCY IPM MEM ETP	CTX-M-15 SHV-1 TEM-1	*tet*(A)
32	Cateter	AMP TZP ATM CZO FEP CAZ CRO CXM CTX CIP MFX NOR GEN TOB SXT TCY IPM MEM ETP	CTX-M-15 SHV-1 TEM-1	*tet*(A)

## Data Availability

The data presented in this study are available on request from the corresponding author.

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
