# Peer review of "TEM,CTX-M,SHV Genes in ESBL-Producing Escherichia coli and Klebsiella pneumoniae Isolated from Clinical Samples in a County Clinical Emergency Hospital Romania-Predominance of CTX-M-15"

_antibiotics, 2022, doi:10.3390/antibiotics11040503_

Round 1

Reviewer 1 Report

The manuscript of Ghenea et al. deals with the antibiotic resistance levels and detection of some antibiotic resistance genes of ESBL E. coli and K. pneumoniae strains from Oltenia region, Romania, The study design and implementation is OK but is only descriptive and specific to a special region that could be done in other parts of Europe with not much novel information.

Special comments

Lanes 27-29: Funny sentence for description of ESBLs, would be better to rephrase.

Lane 31: expensive costs, please, ceheck the English

Lanes 58-59: Please, check the scienetific soundness or the reference of this first sentenece.

Lane 83: instead of 'present' carried by would be a better expression

Table 1: E. coli italic

Lane 183: K. pneumoniae italic

Figure 2: K. pneumoniae italic

Table 2: K. pneumoniae italic

Author Response

Corina Maria Vasile, Phd.

University of Medicine and Pharmacy, Craiova

Craiova, Romania

April 4, 2022

Re: Revision of Manuscript antibiotics-1663789

Dear Academic Editor and Reviewers of the ANTIBIOTICS Journal,

Thank you for your careful consideration of our manuscript and informative comments. Please note below our answers to your comments and let us know if we can address any further comments or provide any additional information for our manuscript to fully meet the ANTIBIOTICS publication criteria.

Best wishes,
Corina Maria Vasile, PhD

Response to Reviewer 1:

Thank you for your valuable comments, which help us to improve our manuscript! Our study even is limited to a specific region, add valuable information about the spread of beta-lactamase genes in Enterobacteriaceae which will be used to develop local antibiotherapy protocols and infection control measures in our hospital.

  1. Reviewer 1: Lanes 27-29: Funny sentence for description of ESBLs, would be better to rephrase.

Author’s response:  Thank you for the observation. We reformulated this sentence.

  1. Reviewer 1: Lane 31: expensive costs, please, check the English

Author’s response:  Thank you for the observation. We reformulated this sentence.

  1. Reviewer 1: Lanes 58-59: Please, check the scientific soundness or the reference of this first sentenece.

Author’s response:  Thank you for the observation. We reformulated this sentence.

  1. Reviewer 1: Lane 83: instead of 'present' carried by would be a better expression

Author’s response:  Thank you for the observation. We changed the word “present” with “existent”.

  1. Reviewer 1: Table 1: E. coli italic

Lane 183: K. pneumoniae italic

Figure 2: K. pneumoniae italic

Table 2: K. pneumoniae italic

Author’s response:  Thank you for the observation. We changed them in italic style.

Reviewer 2 Report

This manuscript by Ghenea et al. reports the prevalence of ESBL genes in E. coli and K. pneumoniae isolates from clinical samples in a county clinical emergency hospital. The authors conclude that blaCTX-M-15 is predominant and a main cause of 3rd generation cephalosporin resistance in E. coli and K. pneumoniae isolates. This reviewer asks the authors to consider the following points.

  1. The authors described that the mcr-1 gene has not been detected in the tested strains. How was the results of colistin susceptibility testings for these strains. Please add.
  2. In some strains (ex. E. coli No.6, K. pneumoniae No.5, No.23….), the 3rd generation cephalosporin resistance determinant has not been shown despite these strains were resistant to ceftazidime and cefotaxime. In addition, some strains appear to be resistant to carbapenems (IPM, MEM, and ETP). These strains may have carbapenemase genes. These should be cleared.

Author Response

Corina Maria Vasile, Phd.

University of Medicine and Pharmacy, Craiova

Craiova, Romania

April 4 , 2022

Re: Revision of Manuscript - antibiotics-1663789

Dear Academic Editor and Reviewers of the ANTIBIOTICS Journal,

Thank you for your careful consideration of our manuscript and informative comments. Please note below our answers to your comments and let us know if we can address any further comments or provide any additional information for our manuscript to fully meet the ANTIBIOTICS publication criteria.

Best wishes,
Corina Maria Vasile, PhD

Response to Reviewer 2:

Thank you for your valuable comments, which help us to improve our manuscript!

  1. Reviewer 2: The authors described that the mcr-1 gene has not been detected in the tested strains. How was the results of colistin susceptibility testings for these strains. Please add.

Author’s response:  Thank you for the observation. We added the observation “all strains were susceptibility to colistin” in the Results section.

  1. Reviewer 2: In some strains (ex. E. coli No.6, K. pneumoniae No.5, No.23….), the 3rd generation cephalosporin resistance determinant has not been shown despite these strains were resistant to ceftazidime and cefotaxime. In addition, some strains appear to be resistant to carbapenems (IPM, MEM, and ETP). These strains may have carbapenemase genes. These should be cleared.

Author’s response:  Thank you for the observation. We added this observation in the Discussion section showing that even if we didn’t tested carbapenemase (KPC, NDMs) genes, they could be present and give also resistance to 3rd generation cephalosporins, even if the genes that encode CTX beta-lactamases are missing, as show studies of substrate activity (citations 76 and 77 added in the text).

Reviewer 3 Report

Dear author,

It is a well written manuscript, it need only few modiffication.

Molecular testing part: please detail the method that was used for the detection of extended spectrum beta-lactamase genes.

please detail the abreviation when are used for the first time in the manuscript.   

Thank you for the opportunity to review this article!

Regards!

Author Response

Corina Maria Vasile,Phd.

University of Medicine and Pharmacy, Craiova

Craiova, Romania

April 4, 2022

Re: Revision of Manuscript - antibiotics-1663789

Dear Academic Editor and Reviewers of the ANTIBIOTICS Journal,

Thank you for your careful consideration of our manuscript and informative comments. Please note below our answers to your comments and let us know if we can address any further comments or provide any additional information for our manuscript to fully meet the ANTIBIOTICS publication criteria.

Best wishes,

Corina Maria Vasile, PhD

Response to Reviewer 3:

Thank you for your valuable comments, which help us to improve our manuscript!

  1. Reviewer 3: Molecular testing part: please detail the method that was used for the detection of extended spectrum beta-lactamase genes.

Author’s response:  Thank you for the observation. We detailed the molecular testing part

  1. Reviewer 3: please detail the abbreviation when are used for the first time in the manuscript.  

Author’s response:  Thank you for the observation. We performed a check through our manuscript and explained correspondingly all abbreviations.

Round 2

Reviewer 2 Report

The revised manuscript could partially response to the comments.

This reviewer hopes that the involvement of other beta-lactamases except CTX-M-15 toward 3rd generation cephalosporin resistance would be completely cleared in the near future study.